# THINKING ISN'T AN ILLUSION: OVERCOMING THE LIMITATIONS OF REASONING MODELS VIA TOOL AUGMENTATIONS

## ABSTRACT

Large Reasoning Models (LRMs) have become a central focus in today's large language model (LLM) research, where models are designed to output a step-by-step thinking process before arriving at a final answer to handle complex reasoning tasks. Despite their promise, recent empirical studies (e.g., [Shojaee et al., 2025] from Apple) suggest that this thinking process may not actually enhance reasoning ability, where LLMs without explicit reasoning actually outperform LRMs on tasks with low or high complexity. In this work, we revisit these findings and investigate whether the limitations of LRMs persist when tool augmentations are introduced. We incorporate two types of tools, Python interpreters and scratchpads, and evaluate three representative LLMs and their LRM counterparts on Apple's reasoning benchmark. Our results show that, with proper tool use, LRMs consistently outperform their non-reasoning counterparts across all levels of task complexity. These findings challenge the recent narrative that reasoning is an illusion and highlight the potential of tool-augmented LRMs for complex reasoning.

## 1 INTRODUCTION

Taking advantage of large-scale pre-training and web-scale training data, Large Language Models (LLMs) (Anthropic, 2024; Achiam et al., 2023; OpenAI, 2024; Llama Team, 2024; McKinzie et al., 2024) have demonstrated unprecedented reasoning capabilities as their model scale continues to increase. This scaling has enabled a range of emergent abilities (Schaeffer et al., 2023; Du et al., 2024), including zero-shot generalization (Brown et al., 2020; Wang et al., 2022) and complex reasoning (Mirzadeh et al., 2025; Wei et al., 2022). Recently, a new class of LLMs, namely Large Reasoning Models (LRMs), has attracted growing interest from the AI community. Models such as OpenAI's o-series (OpenAI, 2024; 2025), DeepSeek-R1 (Guo et al., 2025), and Qwen 3 Thinking (Yang et al., 2025) exhibit significant improvements on various benchmarks compared to non-reasoning LLMs, due to their incorporation of "thinking" strategies such as Chain-of-Thought (CoT) prompting (Wei et al., 2022; Yao et al., 2023; Besta et al., 2024), self-reflection (Ji et al., 2023; Zhao et al., 2025), and process supervision (Chen et al., 2024a; Kim & Suzuki, 2025). These developments suggest a potential paradigm shift in LLMs, where LRMs may represent the next generation of models for complex problem solving.

Despite these advances, there is increasing skepticism about whether LRMs truly improve upon the problem-solving capabilities of conventional LLMs. A number of recent studies have challenged the supposed advantages of LRMs, raising concerns that these models may not possess genuinely deeper reasoning capabilities (Mirzadeh et al., 2025; Shojaee et al., 2025). For example, GSM-Symbolic (Mirzadeh et al., 2025) finds that LRMs tend to rely on pattern matching rather than performing generalizable reasoning. Other studies report that LRMs often generate lengthy outputs filled with redundant tokens that are irrelevant to the final answer (Chen et al., 2024b; Qu et al., 2025; Sui et al., 2025). More notably, Apple's recent "thinking-illusion" benchmark (Shojaee et al., 2025) presents a controlled evaluation comparing LLMs and LRMs across tasks with varying complexity. Their results show that LRMs underperform on simple tasks (e.g., solving the Tower of Hanoi with 4 plates), and fail to show any advantage over LLMs on more complex tasks (e.g., Hanoi with 17 plates), while also consuming significantly more tokens. These findings cast doubt on whether current LRMs offer real improvements in reasoning over standard LLMs.

Figure 1: **Research Question.** Previous empirical results, such as Apple's Thinking-Illusion Benchmark (Shojaee et al., 2025), suggest that Large Reasoning Models (LRMs) do not show clear advantages over standard LLMs when solving complex reasoning problems under controlled problem complexity. In this work, we introduce a new evaluation framework to revisit this conclusion, differing from Apple's setting by allowing LRMs and LLMs to use external tools. We explore whether LRMs exhibit advantages over LLMs in reasoning tasks when tool augmentation is enabled.

However, prior work may have overlooked the fact that benchmark conditions could disadvantage LRMs due to output length limitations. For instance, Apple's benchmark (Shojaee et al., 2025) evaluates reasoning on tasks such as the Tower of Hanoi with up to 20 plates, which may require more than $10^6$ reasoning steps, far exceeding the output token limits of most LLMs (e.g., DeepSeek-R1's limit of 64K tokens (Guo et al., 2025), Qwen 3's limit of 32K tokens (Yang et al., 2025)). As a result, the underperformance of LRMs on hard tasks may not reflect a fundamental reasoning deficiency, but rather an artifact of the limited output window. A natural solution is to augment both models with external tools, such as Python interpreters or scratchpads, to overcome this limitation and better reflect the models' actual reasoning abilities (see Figure 1 for an intuitive illustration). This leads us to the central research question of this paper:

*Under tool augmentation, do LRMs exhibit improved reasoning capabilities compared to LLMs?*

In this paper, we conduct a careful re-examination of the reasoning capabilities of LRMs and LLMs in a tool-augmented setting. Specifically, we equip both models with two basic tools: a Python interpreter and a scratchpad for intermediate computations. We adopt Apple's "thinking-illusion" benchmark (Shojaee et al., 2025) as our evaluation framework, which provides fine-grained control over task complexity and clearly verifiable solutions. Our key distinction from Apple's setting is that we evaluate both LLMs and LRMs under a tool-augmented setup, which the original benchmark does not address. We evaluate two recent LLMs and their corresponding reasoning-augmented variants, and our study reveals several key findings (Section 4):

- We propose a novel LLM evaluation environment by extending the original "thinking-illusion" benchmark (Shojaee et al., 2025) to support tool-augmented evaluation of both LLMs and LRMs. Our framework incorporates a Python interpreter and an innovative scratchpad interface, which mitigates the output length limitations of the original benchmark and enables a fairer and more realistic evaluation of reasoning capabilities.

- We find that with proper tool use, LRMs achieve significant performance improvements on previously unsolvable problems, such as the River Crossing and Blocks World tasks in the Apple's thinking-illusion benchmarks.

- We observe that for certain specific reasoning problems, even with tool use, both LLMs and LRMs still experience notable failures.

- We find that tool use does not necessarily increase token consumption for LRMs.

**Roadmap.** In Section 2, we present our related works. In Section 3, we show our puzzle environments and tool-use settings. In Section 4, we present our main experiment results. In Section 5, we conclude this paper.

## 2  RELATED WORKS

**Fundamental Limits of LRMs.** Despite the progress, recent work has questioned whether LRMs genuinely improve reasoning performance over standard LLMs. Theoretical analyses based on circuit complexity suggest that a Transformer using $k$ CoT steps corresponds to the $\mathsf{TC}^k$ circuit class,

indicating that even multi-step CoT reasoning may be limited in the complexity of problems it can solve (Giannou et al., 2023; Li et al., 2024; Kim & Suzuki, 2025). Empirical evidence also shows that LRMs often generate lengthy outputs with many redundant or irrelevant tokens, increasing inference cost without improving task accuracy (Chen et al., 2024b; Qu et al., 2025; Sui et al., 2025). Furthermore, studies on math reasoning tasks indicate that reinforcement learning may not consistently enhance LRM performance (Mirzadeh et al., 2025). A particularly notable benchmark is Apple's "thinking-illusion" framework (Shojaee et al., 2025), which evaluates both LLMs and LRMs without any tool augmentations under controlled settings with varying task complexities. Their results show that LRMs outperform LLMs only on tasks of medium difficulty, while providing no clear advantage on either simple or very challenging problems.

In this paper, we revisit the evaluation of reasoning capabilities in LLMs and LRMs using a carefully controlled experimental setup. In contrast to previous work (Shojaee et al., 2025), we augment both model types with external tools, specifically a Python interpreter and a scratchpad, and find that LRMs with tool augmentation consistently outperform LLMs with the same tool access. These results challenge prior empirical claims and offer new insights into the potential of LRMs under practical usage scenarios.

Due to space constraints, we move the related works of Large Reasoning Models (LRMs) to Appendix A.1 and LLM Tool Use to Appendix A.2.

## 3 PUZZLE ENVIRONMENTS

In Section 3.1, we introduce our evaluation environment based on Apple's thinking-illusion benchmark. In Section 3.2, we discuss our Python interpreter environments. In Section 3.3, we show the scratchpad tool used in our evaluation.

### 3.1 APPLE'S THINKING-ILLUSION BENCHMARK

To systematically evaluate whether LRMs have improved reasoning capabilities compared to LLMs, we adopt Apple's thinking-illusion benchmark (Shojaee et al., 2025) in our evaluation. While we reuse their puzzle descriptions, we test both LLMs and LRMs in a tool-augmented setup, differing from the original benchmark. This recent benchmark reflects the latest developments in LRM evaluation, offering a controlled, evaluation-friendly environment with clearly defined difficulty levels. Specifically, the benchmark features four types of puzzles that are easy to understand and can be automatically checked using simple verifiers. We evaluate only the correctness of reasoning steps, without considering their optimality, as generating a correct solution alone is already challenging for current LLMs (e.g., in the subtask Checker Jumping, DeepSeek-R1 and DeepSeek-V3 almost fail to solve any problem with $N \geq 3$ in (Shojaee et al., 2025)).

**Hanoi Tower.** This puzzle involves three pillars and $N$ disks, initially placed on the first pillar in descending order (largest at the bottom). The objective is to move all $N$ disks to the third pillar without placing a larger disk on top of a smaller one, and only one disk may be moved per step. The difficulty is directly controlled by the number of disks $N$.

**Checker Jumping.** This is a one-dimensional puzzle with $N$ red checkers on the left, one empty space in the middle, and $N$ blue checkers on the right, placed across $2N + 1$ spaces. The goal is to swap the positions of red and blue checkers, resulting in a mirrored configuration. A checker can move into the adjacent empty space or jump over a checker of the opposite color into the space beyond. The difficulty increases with $N$.

**River Crossing.** This puzzle involves a river with two banks, $N$ actors, and $N$ agents, where each actor is uniquely paired with an agent. Initially, all individuals are on the left bank, and the goal is to move all $2N$ individuals to the right bank. A boat can carry at most $k$ individuals and cannot travel empty. Due to rivalry constraints, actors cannot be left alone with non-paired agents either on the boat or on either bank. We adopt the original setting of $k$ as defined in (Shojaee et al., 2025), and control complexity via $N$.

**Blocks World.** This is a planning puzzle involving three locations and multiple colored blocks. The objective is to rearrange the blocks to achieve a specific color order. Only the topmost blocks at each location can be moved. Task complexity is controlled by the number of blocks $N$.

For all puzzles, we follow the original problem description prompts provided in Section A.1 of (Shojaee et al., 2025) to ensure a fair comparison. To implement each tool-use baseline, we incorporate these prompts into the prompt templates of the corresponding tools.

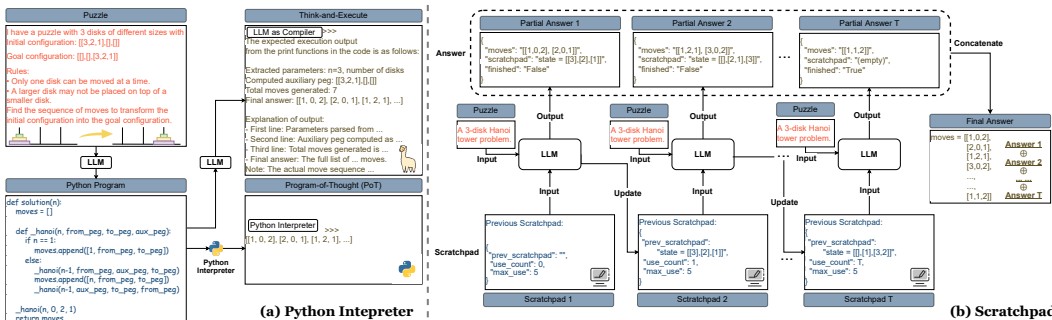

Figure 2: **Evaluation Setting of Tool Use. (a) Python Interpreter:** To address the limited output length issue in LRM evaluation, we introduce two types of Python-based tool usage. The puzzle is first reformulated into Python code by the evaluated LLM, and then executed using either the Program-of-Thought (PoT) or Think-then-Execute framework. In PoT, the external Python interpreter directly executes the code to obtain the answer. In Think-then-Execute, the LLM itself acts as a compiler to interpret the generated code. **(b) Scratchpad:** We use scratchpads as an external memory that allows LLMs and LRMs to store intermediate states and partial answers. This enables models to continue solving the task in multiple steps when the output limit is reached. The LLM determines whether the reasoning process is complete by outputting `Finish:True`. If not finished, the model writes to the scratchpad, which is then fed back along with the puzzle in the next step. The final answer is obtained by concatenating all intermediate partial answers, excluding the scratchpad content itself from the final prediction.

## 3.2 Python Interpreter Environments

Due to the natural output limitations of LRMs and LLMs, they may struggle to solve reasoning problems that require extremely long outputs, such as Blocks World with $N = 13$. A natural solution to enable evaluation on such long-output problems is to incorporate Python code interpreters, which are not subject to the same "memory constraints" as LLMs. In this work, we adopt two Python interpreter-based techniques (Figure 2(a)) to augment LLMs with tools and introduce a new evaluation setting.

**Program-of-Thought (PoT).** Program-of-Thoughts (PoT) (Chen et al., 2023) directly prompts LLMs or LRMs to generate executable Python code, which is then executed using an external Python interpreter. In this paper, we adopt a zero-shot PoT setting without Chain-of-Thought (CoT) reasoning, as the problems are relatively simple in a Python environment (e.g., Hanoi Tower is commonly used as a beginner-level programming exercise). For problem descriptions, we use the same prompts as in (Shojaee et al., 2025), and the prompt template for code generation is adapted from Appendix A.7 of (Chae et al., 2024).

Notably, our evaluation setup differs from (Shojaee et al., 2025) in two key aspects: 1) we incorporate an additional code-generation prompt from (Chae et al., 2024), which is not included in the original thinking-illusion benchmark (Shojaee et al., 2025), and 2) we integrate an external Python interpreter to execute the generated code and produce the final answer.

**Think-and-Execute.** Think-and-Execute (Chae et al., 2024) is a multi-step framework that treats the LLM as a compiler, enabling it to reason directly in the form of Python code. This ensures that the LLM builds a clear abstraction of the puzzle and approaches the solution in a rigorous, structured manner. Specifically, Think-and-Execute begins with a meta prompt that includes a basic system prompt, a task description, and three task-specific examples, guiding the LLM to generate pseudo Python code for execution. The generated code is then interpreted by the LLM itself, rather than by an external Python interpreter, thereby producing an answer through code-like reasoning.

## 3.3 SCRATCHPAD ENVIRONMENTS

Evaluating LRMs and LLMs on problems that require extremely long outputs (e.g., Blocks World with $N = 13$) can be unfair due to the limited output window of these models. Moreover, expecting humans to solve deeply recursive or very long reasoning tasks entirely in their brains is impractical. To address such concerns, we introduce a scratchpad mechanism that provides an external memory for recording intermediate states and allows the model to produce its solution over multiple steps rather than in a single, potentially truncated output. An illustration of our scratchpad environment can be seen in Figure 2(b).

**Key Components.** To ensure the LLM can solve the problem within a limited token budget and avoid infinite loops, we set an upper limit on the number of scratchpad usage steps, denoted as integer $T$. Our scratchpad framework for LLM external memory involves three key components and their interactions:

- **The Large Language Model** $f$: This refers to the evaluated LLM/LRM in our study. We assume that the LLM has no memory of previous dialogue steps in our evaluation, ensuring that we are specifically probing the capabilities of the scratchpad tool-use framework rather than relying on proprietary memory APIs provided by the service.

- **The Partial Answers** $A_1, A_2, \cdots, A_T$: These are the textual outputs generated by the LLM at each step, allowing an overly long answer to be decomposed into manageable segments. For example, when solving a complex Hanoi Tower problem that requires 1023 steps, the LLM can first output the initial 300 steps in $A_1$, the next 300 steps in $A_2$, and the remaining 423 steps in $A_3$. Outputs from different steps are non-overlapping and will be concatenated to form the final answer.

- **The Scratchpad Inputs** $S_1, S_2, \cdots, S_T$: These are textual containers that record the LLM's intermediate reasoning states (e.g., disk placement status in the Hanoi Tower). They can be written arbitrarily by the LLM. At each step $t \in \{1, 2, \cdots, T\}$, the LLM $f$ reads $S_t$ as input and produces $S_{t+1}$ as part of its output.

**Step-wise Prompt with In-Context Examples.** Building on the three key components described earlier, we construct a step-by-step reasoning framework for LLMs using external scratchpad memory. The input prompt to the LLM $f$ at each step consists of three parts. The first part is the puzzle description $P$, which is directly inherited from Apple's Thinking-Illusion benchmark introduced in Section 3.1. The second part is the current scratchpad state $S_t$, which captures intermediate reasoning progress. For the initial step, we use an empty scratchpad $S_1 = \emptyset$.

To ensure the LLM understands how to use the scratchpad, we prepend the prompt with a scratchpad description $D$, which defines the scratchpad interface in a structured JSON format and explains its intended usage. This instruction is shared across all four tasks in the benchmark. Following $D$, we include $m$ in-context examples $\mathcal{E}_m := \{E_1, E_2, \cdots, E_m\}$ tailored to the specific task, demonstrating proper scratchpad usage. Each in-context example includes a task description, an example instance, and the full step-wise output (i.e., both partial answers and scratchpad states) annotated by humans to teach the correct usage pattern.

During inference, the puzzle description $P$, the scratchpad description $D$, and the in-context examples $\mathcal{E}_m := \{E_1, E_2, \cdots, E_m\}$ remain fixed across all steps, while the scratchpad input $S_t$ evolves over time. At each step $t$, the LLM performs the following operations:

$$Z_t = f(P, S_t, D, \mathcal{E}_m),$$
$$A_t, S_{t+1} = \text{Decode}(Z_t),$$

where $Z_t$ is the raw output of the LLM at step $t$, and the partial answer $A_t$ and the updated scratchpad state $S_{t+1}$ are extracted using a decoding function $\text{Decode}(\cdot)$, which can be implemented via simple regex-based parsing or another LLM.

**Final Answer Collection.** After executing all $T$ reasoning steps, we collect the sequence of intermediate partial answers $A_1, A_2, \cdots, A_T$ to form the final output. In our benchmark, where all tasks involve generating a list of simple movement instructions, the final answer can be constructed by applying string-level regex matching and concatenation. For more complex tasks, our framework allows the use of a secondary LLM to aggregate partial outputs into a coherent final answer.

Importantly, executing all $T$ steps is not always necessary. For example, a complex puzzle involving thousands of reasoning steps may require the full sequence, while a simpler task, such as solving the Tower of Hanoi with $N = 3$, can often be completed in a single iteration. To support this flexibility, we enable the LLM to control early stopping. Specifically, at step $t$, if the model outputs `Finish:True` within $A_t$ using the predefined JSON format, we treat $t$ as the early stopping point and skip subsequent steps.

Thus, incorporating optional early stopping, the final answer construction is defined as:

$$T_{\text{final}} := \min\{T,\ T_{\text{cut}}\},$$
$$A_{\text{final}} := \text{Concat}(A_1, A_2, \cdots, A_{T_{\text{final}}}),$$

where $T_{\text{cut}}$ denotes the first step where `Finish:True` is detected. This design enables dynamic reasoning depth and provides a unified framework that balances token efficiency with the ability to perform long-chain reasoning.

## 4 EXPERIMENTS

In Section 4.1, we introduce the main experimental settings of this paper. In Section 4.2, we discuss whether tool use can improve the relative advantage of LRMs compared with LLMs. In Section 4.3, we show several parameter studies.

| Model | Year | Thinking | Output Tokens | # Params |
|---|---|---|---|---|
| DeepSeek-V3 (Liu et al., 2024) | 2024 | No | 8K | 37B |
| DeepSeek-R1 (Guo et al., 2025) | 2025 | Yes | 64K | 37B |
| Qwen 3 (Yang et al., 2025) | 2025 | No | 32K | 32B |
| Qwen 3 Thinking (Yang et al., 2025) | 2025 | Yes | 32K | 32B |

Table 1: **LLM and LRM models evaluated in this paper.**

### 4.1 EXPERIMENTAL SETTINGS

**LLM Models.** In this paper, we primarily use two recent non-thinking LLMs: DeepSeek-V3 and Qwen 3, along with their corresponding long reasoning model (LRM) variants, DeepSeek-R1 and Qwen 3 Thinking. Their basic specifications are summarized in Table 1. We interact with these models through their official APIs for all experiments.

To enable the thinking feature in DeepSeek, one can select the model `"deepseek-reasoner"`. For Qwen 3, thinking is activated by including the flag `"enable_thinking":  true` in the request body. Both DeepSeek and Qwen models require importing the `OpenAI` interface from the `openai` package. To handle possible interruptions in DeepSeek due to long responses, we recommend setting a large timeout parameter (e.g., `timeout=1200`), allowing up to 20 minutes for completion. Each prompt consists of a system prompt and a user prompt. In our setup, the system prompt provides the general problem description, while the user prompt specifies the puzzle instance (e.g., by number). This structure enables easy control over puzzle complexity by adjusting the instance number.

**Parameter Settings.** All experiments in this paper are repeated five times, and we report the average results across runs. For the scratchpad setting, we use $m = 3$ in-context examples, and the maximum number of reasoning steps $T$ is set to $T = 5$. For the number of examples in Think-and-Execute, we follow the official configuration provided by (Chae et al., 2024). For the Program of Thought (PoT) experiments, we use Python version 3.11.13 as the external interpreter.

### 4.2 CAN TOOL USE OVERCOME THE LIMITS OF REASONING MODELS?

In this experiment, we study the core question of this paper: Can tool use help LRMs achieve a performance advantage over standard LLMs? Specifically, we evaluate three tool-use frameworks and compare them against direct prompting (i.e., no tool use). All experiments are repeated 5 times, and we report the number of successful runs out of 5 (i.e., success/trial).

| Tool Usage | LLM Models | Hanoi | | | | | | Checker | | | | | |
|---|---|---|---|---|---|---|---|---|---|---|---|---|---|
| | | N=3 | N=5 | N=7 | N=9 | N=11 | N=13 | N=3 | N=5 | N=7 | N=9 | N=11 | N=13 |
| Direct Prompting | DeepSeek-V3 | 5/5 | 3/5 | 4/5 | 0/5 | 0/5 | 0/5 | 0/5 | 0/5 | 0/5 | 0/5 | 0/5 | 0/5 |
| | DeepSeek-R1 | 5/5 | 5/5 | 5/5 | 0/5 | 0/5 | 0/5 | 0/5 | 0/5 | 0/5 | 0/5 | 0/5 | 0/5 |
| | Qwen 3 | 5/5 | 1/5 | 0/5 | 0/5 | 0/5 | 0/5 | 0/5 | 0/5 | 0/5 | 0/5 | 0/5 | 0/5 |
| | Qwen 3-Thinking | 5/5 | 1/5 | 0/5 | 0/5 | 0/5 | 0/5 | 0/5 | 0/5 | 0/5 | 0/5 | 0/5 | 0/5 |
| Think-and-Execute | DeepSeek-V3 | 5/5 | 5/5 | 5/5 | 0/5 | 0/5 | 0/5 | 0/5 | 0/5 | 0/5 | 0/5 | 0/5 | 0/5 |
| | DeepSeek-R1 | 5/5 | 4/5 | 3/5 | 0/5 | 0/5 | 0/5 | 0/5 | 0/5 | 0/5 | 0/5 | 0/5 | 0/5 |
| | Qwen 3 | 5/5 | 0/5 | 0/5 | 0/5 | 0/5 | 0/5 | 0/5 | 0/5 | 0/5 | 0/5 | 0/5 | 0/5 |
| | Qwen 3-Thinking | 5/5 | 2/5 | 0/5 | 0/5 | 0/5 | 0/5 | 0/5 | 0/5 | 0/5 | 0/5 | 0/5 | 0/5 |
| PoT | DeepSeek-V3 | 5/5 | 5/5 | 5/5 | 5/5 | 5/5 | 5/5 | 0/5 | 0/5 | 0/5 | 0/5 | 0/5 | 0/5 |
| | DeepSeek-R1 | 5/5 | 5/5 | 5/5 | 5/5 | 5/5 | 5/5 | 0/5 | 0/5 | 0/5 | 0/5 | 0/5 | 0/5 |
| | Qwen 3 | 5/5 | 5/5 | 5/5 | 5/5 | 5/5 | 5/5 | 0/5 | 0/5 | 0/5 | 0/5 | 0/5 | 0/5 |
| | Qwen 3-Thinking | 5/5 | 5/5 | 5/5 | 5/5 | 5/5 | 5/5 | 0/5 | 0/5 | 0/5 | 0/5 | 0/5 | 0/5 |
| Scratchpad | DeepSeek-V3 | 5/5 | 5/5 | 2/5 | 0/5 | 0/5 | 0/5 | 0/5 | 0/5 | 0/5 | 0/5 | 0/5 | 0/5 |
| | DeepSeek-R1 | 5/5 | 5/5 | 3/5 | 0/5 | 0/5 | 0/5 | 0/5 | 0/5 | 0/5 | 0/5 | 0/5 | 0/5 |
| | Qwen 3 | 5/5 | 1/5 | 0/5 | 0/5 | 0/5 | 0/5 | 0/5 | 0/5 | 0/5 | 0/5 | 0/5 | 0/5 |
| | Qwen 3-Thinking | 5/5 | _2/5_ | 0/5 | 0/5 | 0/5 | 0/5 | 0/5 | 0/5 | 0/5 | 0/5 | 0/5 | 0/5 |

Table 2: **Accuracy results for Hanoi Tower and Checker Jumping**. Cases that LRMs outperform LLMs are denoted by underline.

| Tool Usage | LLM Models | River | | | | | | Block | | | | | |
|---|---|---|---|---|---|---|---|---|---|---|---|---|---|
| | | N=3 | N=5 | N=7 | N=9 | N=11 | N=13 | N=3 | N=5 | N=7 | N=9 | N=11 | N=13 |
| Direct Prompting | DeepSeek-V3 | 0/5 | 0/5 | 0/5 | 0/5 | 0/5 | 0/5 | 4/5 | 3/5 | 0/5 | 0/5 | 0/5 | 0/5 |
| | DeepSeek-R1 | 1/5 | 0/5 | 0/5 | 0/5 | 0/5 | 0/5 | 5/5 | 4/5 | 0/5 | 0/5 | 0/5 | 0/5 |
| | Qwen 3 | 0/5 | 0/5 | 0/5 | 0/5 | 0/5 | 0/5 | 5/5 | 0/5 | 0/5 | 0/5 | 0/5 | 0/5 |
| | Qwen 3-Thinking | 0/5 | 0/5 | 0/5 | 0/5 | 0/5 | 0/5 | 5/5 | 3/5 | 2/5 | 0/5 | 0/5 | 0/5 |
| Think-and-Execute | DeepSeek-V3 | 0/5 | 0/5 | 0/5 | 0/5 | 0/5 | 0/5 | 5/5 | 5/5 | 0/5 | 0/5 | 0/5 | 0/5 |
| | DeepSeek-R1 | 0/5 | 0/5 | 0/5 | 0/5 | 0/5 | 0/5 | 5/5 | 4/5 | _2/5_ | _2/5_ | _1/5_ | _1/5_ |
| | Qwen 3 | 0/5 | 0/5 | 0/5 | 0/5 | 0/5 | 0/5 | 5/5 | 4/5 | 2/5 | 0/5 | 0/5 | 0/5 |
| | Qwen 3-Thinking | 0/5 | 0/5 | 0/5 | 0/5 | 0/5 | 0/5 | 5/5 | 3/5 | 1/5 | 0/5 | 0/5 | 0/5 |
| PoT | DeepSeek-V3 | 0/5 | 0/5 | 0/5 | 0/5 | 0/5 | 0/5 | 1/5 | 1/5 | 1/5 | 1/5 | 1/5 | 1/5 |
| | DeepSeek-R1 | _4/5_ | _4/5_ | _4/5_ | _4/5_ | _4/5_ | _4/5_ | _5/5_ | _5/5_ | _5/5_ | _5/5_ | _5/5_ | _5/5_ |
| | Qwen 3 | 0/5 | 0/5 | 0/5 | 0/5 | 0/5 | 0/5 | 2/5 | 2/5 | 2/5 | 2/5 | 2/5 | 2/5 |
| | Qwen 3-Thinking | 0/5 | 0/5 | 0/5 | 0/5 | 0/5 | 0/5 | _5/5_ | _5/5_ | _5/5_ | _5/5_ | _5/5_ | _5/5_ |
| Scratchpad | DeepSeek-V3 | 0/5 | 0/5 | 0/5 | 0/5 | 0/5 | 0/5 | 5/5 | 1/5 | 0/5 | 0/5 | 0/5 | 0/5 |
| | DeepSeek-R1 | _1/5_ | 0/5 | 0/5 | 0/5 | 0/5 | 0/5 | _5/5_ | _5/5_ | 3/5 | _4/5_ | _4/5_ | 0/5 |
| | Qwen 3 | 0/5 | 0/5 | 0/5 | 0/5 | 0/5 | 0/5 | 5/5 | 4/5 | 5/5 | 0/5 | _0/5_ | 0/5 |
| | Qwen 3-Thinking | 0/5 | 0/5 | 0/5 | 0/5 | 0/5 | 0/5 | 5/5 | 1/5 | 0/5 | 0/5 | 0/5 | 0/5 |

Table 3: **Accuracy results for River Crossing and Blocks World**. Cases that LRMs outperform LLMs are denoted by underline.

We focus on the four subtasks in Apple's Thinking Illusion benchmark (described in Section 3.1) and vary the task complexity using a parameter $N \in 3, 5, 7, 9, 11, 13$. Simpler cases with $N \leq 2$ are omitted because almost all models perform well in those settings.

Results for the Hanoi Tower and Checker Jumping tasks are shown in Table 2, and results for River Crossing and Blocks World are shown in Table 3. We highlight four key observations:

**1) Tool use with Program of Thought (PoT) enables major improvements for LRMs on multiple tasks.** We find that PoT significantly boosts the performance of LRMs like DeepSeek-R1. For example, on both River Crossing and Blocks World, DeepSeek-R1 achieves around 80% accuracy with tool use, while its non-thinking variant DeepSeek-V3 fails to solve most cases. On River Crossing, DeepSeek-V3 performs near zero, while DeepSeek-R1 with PoT achieves consistent success. Similarly, for Blocks World, PoT lifts accuracy from 20% to 80%. Another striking result is observed in Hanoi Tower: previously, both DeepSeek-V3 and DeepSeek-R1 struggled with large $N$, but with PoT, they achieve perfect accuracy due to the structured nature of the problem, which is well-suited for external Python programs.

**2) Some hard problems remain unsolved even with tool use.** Checker Jumping remains unsolved for $N \geq 3$ across all models and tool-use methods. This aligns with Apple's original benchmark, where only $N = 1$ and $N = 2$ are solvable. This result suggests that while tools help in many cases, there are still hard reasoning tasks that remain out of reach.

**3) The effectiveness of tool use depends on the base model's strength.** Not all models benefit equally from tool use. While DeepSeek-R1 shows strong improvements with PoT and Scratchpad (e.g., on River Crossing and Blocks World), Qwen-3 shows only limited gains, mostly on Blocks

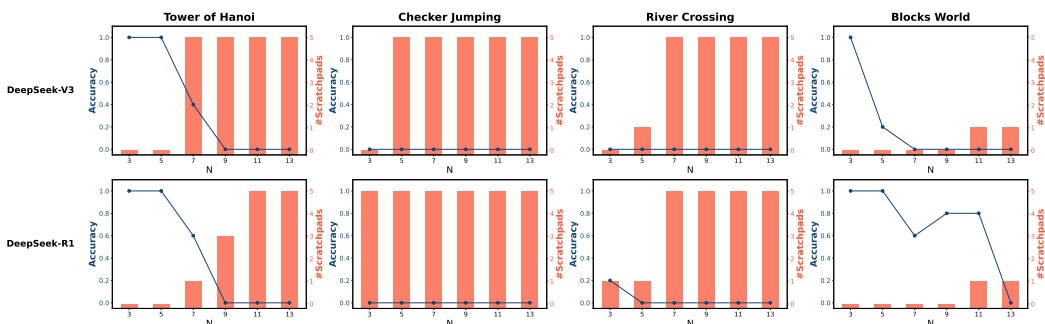

Figure 3: **Number of Scratchpads Used on DeepSeek-V3 and DeepSeek-R1.**

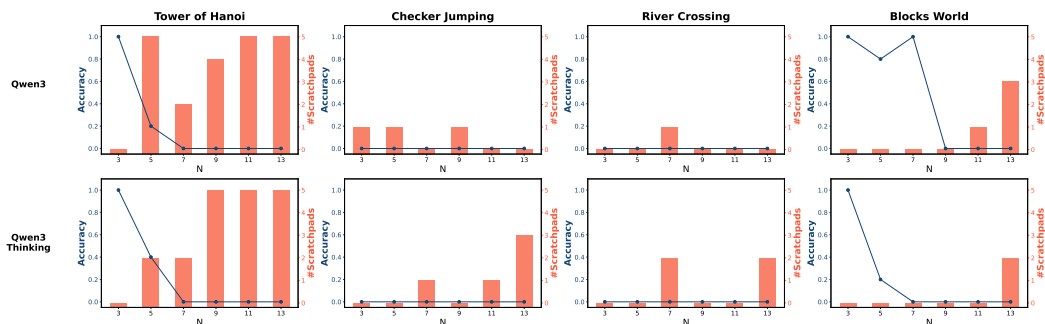

Figure 4: **Number of Scratchpads Used on Qwen 3 and Qwen 3 Thinking.**

World with PoT. In general, Qwen 3 also performs worse across tasks. This suggests that tool use helps only when the base model is strong enough to utilize the tools effectively.

**4) Some tool-use frameworks are more effective than others.** Among the three tool-use methods, PoT delivers the strongest improvements. Scratchpad also helps significantly, especially on Blocks World from $N = 3$ to $N = 11$, where accuracy jumps from 0–20% to 60–100%. In contrast, Think-and-Execute provides minimal gains. The only notable case is DeepSeek-R1 on Blocks World from $N = 7$ to $N = 13$, where accuracy increases by only 20–40%. Overall, PoT is the most powerful tool framework, followed by Scratchpad, with Think-and-Execute being the least effective in our experiments.

### 4.3 HYPERPARAMETER STUDIES

**Scratchpad Chain Length.** In this parameter study, we examine whether task complexity is directly related to the number of scratchpads used for reasoning in our scratchpad-based tool-use framework, as described in Section 3.3. Specifically, we record how many times the model invokes the scratchpad and requests a pause due to long outputs before continuing to the next reasoning step. The results for DeepSeek-V3 and DeepSeek-R1 are shown in Figure 3, and the results for Qwen-3 and Qwen-3-Thinking are shown in Figure 4. In these figures, model accuracy is plotted as a line, and the number of scratchpad invocations is shown as bars.

First, we observe that different base models exhibit highly distinct patterns of scratchpad usage. For example, DeepSeek models tend to use the maximum allowed number of 5 scratchpads on the Checker and River Crossing problems, while the Qwen models typically use only 0–2 scratchpads on the same tasks. Second, the number of scratchpad invocations is similar between reasoning and non-reasoning variants of each model, showing no significant difference. This suggests that both types of models have a similar tendency to use the scratchpad tool, and that the key differences lie in the base model architecture rather than the presence of explicit reasoning instructions.

**Token Consumption.** In this study, we investigate whether the use of external tools increases the number of tokens consumed by reasoning models. Specifically, we evaluate Qwen 3 Thinking

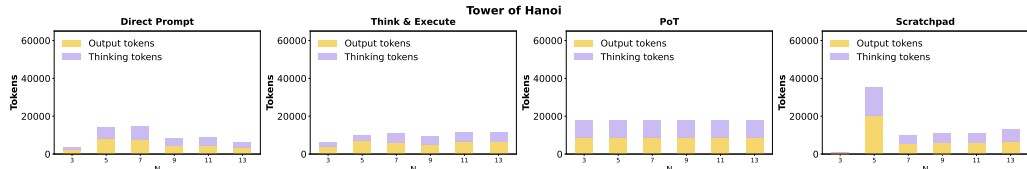

Figure 5: **Token Consumption of Qwen 3 Thinking on Tower of Hanoi**.

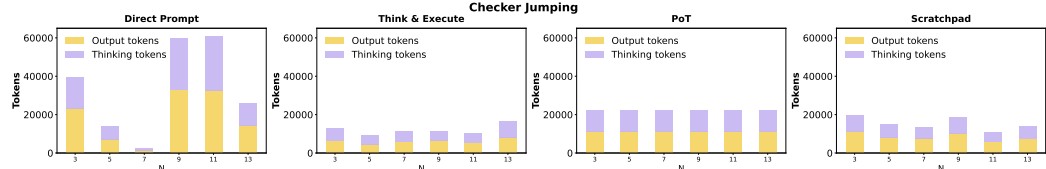

Figure 6: **Token Consumption of Qwen 3 Thinking on Checker Jumping**.

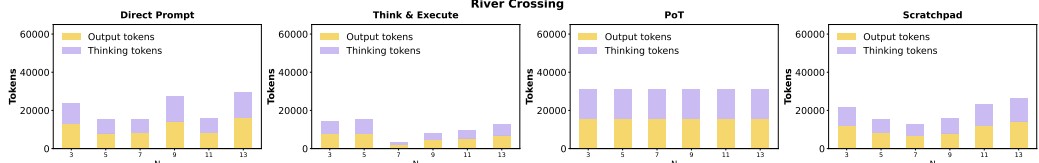

Figure 7: **Token Consumption of Qwen 3 Thinking on River Crossing**.

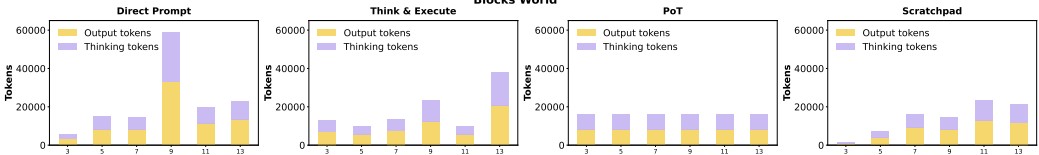

Figure 8: **Token Consumption of Qwen 3 Thinking on Blocks World**.

across all four tool-use baselines and four datasets from Apple's benchmark. The results are shown in Figures 5–8. First, we observe that tool-use frameworks involving multi-step reasoning, such as Scratchpad or Think-and-Execute, do not necessarily lead to higher token consumption. This advantage is especially evident on the Checker Jumping and Blocks World tasks. Second, when analyzing the composition of the freed tokens, we find that both the thinking tokens and the output tokens are reduced, without any particular bias toward one type. This may be because tool use helps guide the model toward more effective reasoning paths, allowing it to avoid unnecessary or unproductive trials.

## 5 CONCLUSION

This work revisits the reasoning capabilities of Large Reasoning Models (LRMs) by introducing external tools such as Python interpreters and scratchpads. Contrary to prior studies suggesting LRMs offer limited benefit over standard LLMs, our results show that tool-augmented LRMs consistently outperform their non-reasoning counterparts, especially on previously unsolvable reasoning tasks. These findings challenge the recent skepticism around LRMs and underscore the importance of tool augmentation when evaluating model reasoning.

Looking forward, several promising directions remain open. First, extending tool use beyond basic interpreters to more structured environments (e.g., symbolic solvers or simulators) may further unlock the reasoning potential of LRMs. Second, understanding the failure modes of both LLMs and LRMs under complex tool-based workflows is essential for robustness. Lastly, future benchmarks should incorporate tool interactions as a first-class component to better reflect real-world problem-solving scenarios.

## ETHIC STATEMENT

This paper does not involve human subjects, personally identifiable data, or sensitive applications. We do not foresee direct ethical risks. We follow the ICLR Code of Ethics and affirm that all aspects of this research comply with the principles of fairness, transparency, and integrity.

## REPRODUCIBILITY STATEMENT

We ensure the reproducibility of our empirical findings. For all experiments, we describe the sources of the LLM models, datasets, and API settings in the main text. All prompt templates used are also provided to support the reproducibility of our results.

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

# Appendix

**Roadmap.**  In Section A, we present our addtional related works.

## A  ADDITIONAL RELATED WORKS

In Section A.1, we present the related works on large reasoning models (LRMs). In Section A.2, we show the related works on tool usage in LLMs.

### A.1  LARGE REASONING MODELS (LRMS)

Reasoning is a core capability of intelligent autonomous systems. Recent advances in Large Language Models (LLMs) have demonstrated that scaling up training data and model size significantly enhances general capabilities, including emergent properties such as zero-shot generalization (Brown et al., 2020; Wang et al., 2022) and logical reasoning (Mirzadeh et al., 2025; Wei et al., 2022), as evidenced by the success of many commercial models (Anthropic, 2024; Achiam et al., 2023; OpenAI, 2024; Llama Team, 2024; McKinzie et al., 2024). A natural follow-up question is whether the step-by-step thinking process of LLMs can also be explicitly supervised (OpenAI, 2024; 2025; Chen et al., 2024a), enabling more structured, human-like task decomposition and deeper reasoning. This is often referred to as scaling the model's test-time computation (Snell et al., 2025; Liu et al., 2025). These efforts have led to notable progress in domains such as mathematical reasoning (Mirzadeh et al., 2025) and logical problem solving (Parmar et al., 2024), giving rise to a new class of models known as Large Reasoning Models (LRMs).

LRMs are designed with specific techniques to support more structured reasoning. For example, they extend earlier prompting methods such as Chain-of-Thought (CoT) (Wei et al., 2022; Yao et al., 2023; Besta et al., 2024) with additional capabilities like self-verification, enabled through high-quality CoT supervision from human experts. However, such expert annotations are expensive and limited in scale. As a result, a growing body of work explores the use of reinforcement learning (RL) to generate reasoning trajectories without direct supervision (Zelikman et al., 2022; Goyal et al., 2024; Shao et al., 2024). This has been shown to be a viable alternative in several reasoning-focused models, such as DeepSeek-R1 (Guo et al., 2025).

### A.2  LLM TOOL USE

Due to inherent limitations in Large Language Models (LLMs), such as restricted output length and hallucinations (Ji et al., 2023; Chen et al., 2024c), a growing body of research has explored the use of external tools to enhance their problem-solving capabilities. Early studies focused on integrating a single tool with LLMs, including search engines (Shuster et al., 2022), web browsers (Nakano et al., 2021), Python interpreters (Chen et al., 2023; Chae et al., 2024), calculators (Thoppilan et al., 2022), and external memory buffers (e.g., scratchpads) (Zhong et al., 2024). For example, Program-of-Thought (PoT) (Chen et al., 2023) allows an LLM to first generate Python code, which is then executed by an external interpreter to obtain the final result. Subsequent work has developed unified frameworks that allow LLMs to interact with a wide range of tools through standardized APIs (Schick et al., 2023; Zhuang et al., 2023). These advances have also led to the emergence of LLM agent systems (Zhao et al., 2024; Wang et al., 2024), in which LLMs are equipped with tools and capable of interacting with other LLMs to solve complex tasks collaboratively. In this paper, we revisit prior empirical findings that suggest LRMs do not show significant advantages over LLMs when using tools. By systematically evaluating LRMs and LLMs under a tool-augmented setting, we demonstrate that tool use can unlock the reasoning potential of LRMs, revealing clear benefits that were overlooked in earlier benchmarks.

## LLM USAGE DISCLOSURE

LLMs were used only to polish language, such as grammar and wording. These models did not contribute to idea creation or writing, and the authors take full responsibility for this paper's content.

