# OpenReview forum: "Thinking Isn't an Illusion: Overcoming the Limitations of Reasoning Models via Tool Augmentations"
_ICLR.cc/2026/Conference — ICLR 2026 Conference Withdrawn Submission_

### Official Review · Reviewer_EpLK · 2025-10-18

**Soundness:** 1
**Presentation:** 3
**Contribution:** 1
**Rating:** 2
**Confidence:** 5

**Summary:**

This paper challenges the position put forward by [Shojaee et al, 2025] and argues that Large Reasoning Models (LRMs) can outperform standard Large Language Models (LLMs) augmented with external tools on benchmarks featuring dynamic computational complexity.

**Strengths:**

The paper is generally well-written and easy to follow. The experimental setup and key findings are clearly presented, and the design choices appear reasonable. The authors conduct controlled ablations between reasoning and non-reasoning models, use benchmarks with well-defined complexity (directly adapted from [Shojaee et al., 2025] though), and incorporate practical tools such as a Python interpreter and scratchpad. Overall, these settings provide a reasonable foundation for the study’s discussion.

**Weaknesses:**

W1. Misinterpretation of prior work (Shojaee et al., 2025).

The paper partially misinterprets and oversimplifies the conclusions of Shojaee et al. (2025). In that work, the authors did not primarily focus on comparing standard LLMs and LRMs. Rather, their key claim was that the reasoning traces produced by LRMs are not consistently reliable across tasks with varying computational complexity—LRMs tend to perform well at medium-level complexity, worse than standard LLMs on low-complexity tasks, and completely fail on high-complexity ones. The current paper shifts this focus to the statement that “standard LLMs outperform LRMs on tasks with low or high complexity,” which does not accurately capture the intent of Shojaee et al. (2025). Furthermore, the new results in Tables 2 and 3 still show that neither reasoning models nor standard LLMs can consistently overcome the challenges of complexity scaling across various tasks.

W2. Limited depth of analysis and insufficient evidence for claims.

The analysis and findings presented in the paper are not sufficiently in-depth to substantiate its central claims. While the authors aim to demonstrate that explicit reasoning procedures enhance problem solving—particularly when tools are integrated into the chain-of-thought—the experimental results indicate only marginal gains in most cases (Tables 2 and 3), and in some settings, reasoning models even underperform the standard LLM baseline (e.g., the Hanoi task with N = 5 and N = 7, Think-and-Execute, DeepSeek-R1 vs. DeepSeek-V3). Moreover, the paper does not convincingly explain how reasoning contributes to improved problem solving: the tool-use frequencies shown in Figure 4 are nearly identical between reasoning and non-reasoning conditions. Consequently, readers are left without clear insights into why or when reasoning models offer tangible advantages in handling complex problems.

W3. Lack of discussion on inconsistent task performance.

Across the four evaluated tasks, models exhibit quite different problem-solving curves as task complexity grows. However, the paper provides very limited analysis or interpretation of these inconsistencies between tasks. This lack of analysis weakens the overall contribution, as readers gain little understanding of why performance diverges across tasks or what factors drive these differences.

Overall, the work lacks in-depth experimental analysis and primarily reports results of tool-augmented reasoning on an existing benchmark. As a result, its contribution remains limited. To strengthen the paper, the authors should perform a deeper investigation into how reasoning mechanisms fundamentally benefit problem solving, thereby providing concrete evidence for their central claim that “thinking isn’t an illusion” or they can “overcome the limitations of reasoning models” (See also Q1–Q3 for related points.)

**Questions:**

Q1. It remains unclear why the performance varies so drastically across different tasks. For example, in the Hanoi task with coding tools (PoT), even non-reasoning models perform well at large problem complexity (e.g., N = 13), whereas all experiments on the Checker problem fail completely. What distinguishes these two problem settings that leads to such a large performance gap? A discussion of the task-specific characteristics that contribute to this discrepancy would be helpful.

Q2. What exactly helps problem solving?  For PoT, the model must write executable code to solve problems. How does reasoning improve the quality of this generated code? Does it help the model handle corner cases more effectively, or encourage self-reflection and correction when execution errors occur? Such analysis is crucial to understanding how reasoning contributes to problem solving, but is missing from the current discussion.

Q3. Figures 3 and 4 (and Line 428) reveal that reasoning and non-reasoning models exhibit similar tool-calling frequencies, yet their problem-solving performance differs. What accounts for this discrepancy? Additional analysis would help clarify whether the reasoning process affects how the tools are used.

Q4. Since the paper is titled "overcoming the limitations of reasoning models", can you summarize concisely: what exactly are the limitations of reasoning models?  How does tool augmentations overcome them?

---

> ### Author Response · Authors · 2025-11-27
>
> Thank you for your thoughtful feedback. Your comments are very helpful and much appreciated. We will address these in the next version.

---

### Official Review · Reviewer_v5i1 · 2025-10-26

**Soundness:** 1
**Presentation:** 2
**Contribution:** 2
**Rating:** 2
**Confidence:** 4

**Summary:**

This paper challenges the claim by Shojaee et al. (2025) and argues that under the tool augmentation setting, the reasoning models consistently outperform the non-reasoning counterparts across all levels of task complexity. To prove that, they first make an evaluation environment based on the previous work's benchmark, and then test model's ability to solve problems using Python interpreter or scratchpad.

**Strengths:**

1. Authors point out an interesting point that is not covered in the apple's thinking-illusion benchmark, and show that LLM with tool usage could weaken the previous work's claim.
2. The paper is easy to read and follow, and the contents are well-organized by each (sub)section.
3. If the author's claim is true, then it will be a big contribution to the community to understand the reasoning model's behavior.

**Weaknesses:**

1. The argument in a paragraph in line 68-77 is not convincing to me. It needs more evidence and justifications to be logically sound.
1.1 Token limit constraint is equally applied to both LLMs and LRMs, but can we argue that only LRM's performance is mistakenly measured? (Couldn't LLM's performance be increased more when there is no constraint in token limit?)
1.2 What is the actual percentage of LRM failure leaded by the token limit? -- If this percentage is low, then it is hard to believe the paper's arguments.
1.3 I don't feel "Augmenting models with external tools" is the only natural solution. How many tokens are saved by introducing the tools? Is there any other effective natural solution better than tool augmentation? What if we summarize the model's reasoning if the token limit is reached, and then fed it to the model to continue its reasoning? (I would happy to hear detailed justifications by authors.)

2. In this paper's setting, models are called several times for tool use. is it a fair correspondence with the setting in Shojaee et al. (2025)? if the token limit could be resolved by calling the model several times, then do we necessarily need tools to overcome the token limitation?

3. Authors often use LLMs as both reasoning & non-reasoning models or non-reasoning models alone. This should be clarified in the paper.

4. I'm not sure the way authors show results in Table 2 and 3 would be the best way. It's hard to connect the main claim of the paper with the table. Furthermore, the paper's main claim is LRM outperforms LLM in a tool use setting. However, the tables' results do not strongly support this claim. (there is no such clear trend.)

**Questions:**

See Weaknesses.

---

> ### Author Response · Authors · 2025-11-27
>
> Thank you for your thoughtful feedback. Your comments are very helpful and much appreciated. We will address these in the next version.

---

### Official Review · Reviewer_gaxT · 2025-10-31

**Soundness:** 1
**Presentation:** 3
**Contribution:** 1
**Rating:** 2
**Confidence:** 3

**Summary:**

This paper presents empirical results showing that large reasoning models (LRM), when equipped with tool usage capabilities (or "tool augmentations"), is able to outperform plain LLMs without reasoning capabilities. Specifically, the paper proposes an evaluation environment where LRMs are able to access a Python interpreter and a scratchpad environment. The LRM and LLMs are tested on four tasks, namely, Hanoi tower, checker jumping, river crossing, and blocks world, with tools usages are provided in four modes (direct prompt, think-and-execute, program of thought, and with scratchpad).

Empirical results on DeepSeek-V3/R1 and Qwen-3/3 Thinking demonstrate that LRMs' performance are not necessarily poorer than general-purpose LLMs when tools are available. Furthermore, tools usage does not necessarily results in higher token usage.

**Strengths:**

1. The paper provided empirical evidences for a problem that is of interest to the community: comparing performance of reasoning and non-reasoning LLMs.
2. The paper is well written and results are clearly presented. Overall the message conveyed by this paper is clear (though I do not believe it has been proven to be generalisable or of significant technical depth).

**Weaknesses:**

1. The experiment does not cover enough models. Only DeepSeek and Qwen-3 LLM families are tested. It is unclear to me whether the result is applicable to other reasoning models, especially close-weight models such as Gemini, GPT, and Claude.

2. The technical contribution of proposed benchmark is weak. Tool-augmented LLMs have long been deployed a long time and a fair amount of empirical evidence has proven that tool usage improves model performance. So the results in this paper does not look surprising to me.

**Questions:**

1. What are the capabilities/characteristics of LRMs that really makes them outperform LLMs with tool usage? The empirical results in this paper do not answer this question. Without answering this question, it is difficult to generalise results on specific models to all reasoning models.

2. How are the results in this paper proving that "thinking is not an illusion"?

**Details Of Ethics Concerns:**

None.

---

> ### Author Response · Authors · 2025-11-27
>
> Thank you for your thoughtful feedback. Your comments are very helpful and much appreciated. We will address these in the next version.

---

### Official Review · Reviewer_XAxz · 2025-10-31

**Soundness:** 3
**Presentation:** 2
**Contribution:** 2
**Rating:** 4
**Confidence:** 5

**Summary:**

This paper challenges recent claims that reasoning in Large Reasoning Models (LRMs) is an "illusion" by reevaluating their performance under tool-augmented settings. Using Python interpreters and scratchpads, the authors demonstrate that LRMs consistently outperform standard LLMs across various task complexities in Apple’s thinking-illusion benchmark.

**Strengths:**

1.  The paper provides a timely and relevant rebuttal to the "thinking as illusion" narrative, supported by experimentation.
2. The introduction of a scratchpad mechanism and Python interpreter integration offers a practical and scalable approach to evaluating long-horizon reasoning.

**Weaknesses:**

1. Insufficient Experimental Analysis: The results are largely descriptive and lack in-depth analysis—e.g., why certain tools work better than others, or why some tasks remain unsolvable.
2. Writing Quality: While understandable, the writing is often dense and could benefit from clearer transitions, better structuring, and more engaging engaging exposition.

**Questions:**

1. The authors argue that the underperformance of LRMs on difficult tasks is merely an artifact caused by limited output windows, thus necessitating the augmentation of both models with external tools. They further analyze the reasoning capabilities of LRMs in this tool-augmented setting. Overall, I find this a very interesting hypothesis. However, your work is solely based on the "thinking-illusion" benchmark, which I believe is insufficient for a comprehensive evaluation. Validation across more datasets is needed.
2. Similar to the first question, I find the overall experimental analysis in this paper to be insufficient (despite the interesting motivation). For instance, the number of tested LLMs and LRMs is too small, and the analysis lacks depth.
3. This paper seems more akin to a benchmark study. I wonder if your current selection for the "Primary Area" option might be inappropriate.
4. There are also some writing issues in the paper; for example, the formula on line 259 is misaligned.
5. I suggest analyzing the performance of LLMs and LRMs with varying parameter sizes within your proposed framework.

---

> ### Author Response · Authors · 2025-11-27
>
> Thank you for your thoughtful feedback. Your comments are very helpful and much appreciated. We will address these in the next version.

---

### Note · Authors · 2025-11-27

**Comment:**

We would like to sincerely thank all the reviewers for providing insightful feedback. After careful consideration, we have decided to withdraw this paper.

**Withdrawal Confirmation:**

I have read and agree with the venue's withdrawal policy on behalf of myself and my co-authors.